# Agronomical and Physiological Responses of Faba Bean Genotypes to Salt Stress

**Muhammad Afzal** [1] , **Salem S. Alghamdi** [1] , **Hussein H. Migdadi** [1,2,*] , **Ehab El-Harty** [1] and **Sulieman A. Al-Faifi** [1]

1   Department of Plant Production, College of Food and Agriculture Sciences, King Saud University, Riyadh 11451, Saudi Arabia; mmushtaq@ksu.edu.sa (M.A.); salem@ksu.edu.sa (S.S.A.); ehabelharty@gmail.com (E.E.-H.); salfaifi@ksu.edu.sa (S.A.A.-F.)
2   National Agricultural Research Center, Baqa, Amman 19381, Jordan
*   Correspondence: hmigdadi@ksu.edu.sa; Tel.: +966-535-871-345

**Abstract:** Considering the importance of salinity stress and genotype screening under stress conditions, the current study evaluated faba bean genotypes in response to saline stress and identified those that were tolerant and determined the influential ratio of each yield component on seed yield under both conditions. As a result, 12 faba bean genotypes were tested under 2 levels of salt stress (100 mM and 200 mM) and a control. The study was analyzed with multivariate (descriptive, ANOVA, PCA, biplot, cluster analysis, and indices) analysis techniques to determine the tolerance level of each genotype. Similarly, the cluster analysis results reported that faba bean genotypes were divided into two groups under the control and 100 mM salinity levels; however, the 200 mM salinity level recorded three groups of faba bean genotypes, showing that salinity stress may limit phenotypic variability among faba bean genotypes. The descriptive analysis results showed a wide range of diversity among the studied characteristics under control and salinity stress conditions. The number of seeds/plants recorded a significant association with plant height (cm) (PH), stomatal conductance (SC), days to flowering (DF), the number of pods, and seed weight (g) (SW); however, an insignificant association was recorded with leaf temperature (LT), fresh weight (g) (FW), $Na^+$, $K^+$, and Na/K ratio. The first three principal components (PCs) represent 81.45% of the variance among the studied traits. The most significant characteristics that contributed the most to the diversity were (PH, leaf area, SPAD reading, stomatal conductance, DF, number of pods/plants, number of seeds/pods, SW, K, and total chlorophyll content); however, the significant genotypes (Hassawi-2, Sakha, ILB-4347, Misr-3, FLIP12501FB) were present in PC1 under both conditions. The results predicted that Hassawi-2, ILB-4347, Sakha, Misr-3, and Flip12501FB were the significant (tolerant) genotypes. However, FLIP12504FB represents a sensitive genotype based on its final grain yield. The results of the indices also recorded significant index correlations with grain yield, demonstrating that these indices are effective tools for screening faba bean-tolerant genotypes under salinity stress conditions.

**Keywords:** salinity; abiotic stress; PCA; indices; *Vicia faba*

## 1. Introduction

Soil salinization is one of the most critical abiotic stressors that impacts crop yields globally; salinity threatens roughly 6% of the world's total land area, including 20% of arable land and 33% of irrigated land [1]. Land salinization is increasing, with 10 million acres of agricultural land damaged each year by salt buildup caused by human activities and other climate change-related variables [2]. Plant growth and productivity are dramatically reduced by salinity stress, which can drastically affect production [3]. Faba bean is one of the oldest crops in the world and its cultivation dates back to the Mediterranean region [4]. Globally, 4.7 million tons of grain legume crop were produced over 3.4 million hectares [5]. Despite its age and commercial importance, the fact is that faba bean is a

diploid species (2n = 12), and modest progress has been made in developing an excellent genetic understanding of this crop. Faba bean has an exceptionally large genome of approximately 13.4 Gb [6], and it considered to be the largest genome in the grain legume family. *Vicia faba* L. is a cool-season legume crop and produces high-protein grains for human production and livestock in developing regions [7,8]. The Vicia genus belongs to the *Viceae* tribe, a cool-season clade of the subfamily. Papilionoideae is part of the legume family Fabaceae [9]. Because of the increasing demand for faba bean consumption in the middle east, there is a need to develop faba bean genotypes that are suitable for arid and semiarid regions [8].

Salinity tolerance is a complicated physiological feature with several sub-components; the traditional view holds that salinity impacts plant performance through osmotic stress and specific ion toxicity [10]. Salt concentration in soil severely affects faba bean yield [11]; however, faba bean seed germination is more sensitive to growth under salinity. Characterizing salt tolerance, faba bean genotypes have shown enhanced productivity under salinity [12]. Plants compensate for decreasing osmotic potential in the rhizosphere by minimizing water loss. It has been claimed that stomatal transpiration accounts for around 95% of plant water loss [13]. Lower stomatal density is an essential physiological characteristic in salinity-tolerant quinoa [14,15], while farmed barley uses a stress-escaping strategy by lowering stomatal density to conserve water when grown in saline circumstances [16]. Saline-induced stomatal closure would reduce $CO_2$ inflow, reducing leaf photosynthetic capacity and, eventually, yield [17]. Ionic toxicity, which is induced by high sodium accumulations in the cytoplasm, is fundamental for plants under salt stress [18]. $Na^+$ exclusion from the shoot is thought to be crucial for plants to overcome the adverse effects of increasing salt, and a large percentage of $Na^+$ exclusion (>98%) in wheat is achieved by limiting net $Na^+$ absorption at the soil–root interface and net xylem loading in roots [19]. The $K^+/Na^+$ ratio is thought to be the fundamental trait imparting salinity stress resistance in plants, and it is frequently used as a screening tool for plant breeders [20]. Under salt stress, $Na^+$ and $K^+$ transporters are critical for maintaining $Na^+$ and $K^+$ homeostasis in cells and plants [21]. Enhancing the salt-tolerance capacity is one of the most efficient and viable approaches for reducing the negative impact that salinity has on crop output [22].

Many screening techniques for salt tolerance have been conducted. The replicability of the experiments and the consistent result amongst laboratories remains challenging because of a lack of a standard growing environment [23], and a few of these experiments were part of large-scale studies [24,25]. Even though salinity tolerance is a polygenic trait, several studies have treated it as a single trait and have evaluated it using visual scoring [26]. The pyramiding of favorable morphological, physiological, and biochemical factors can enhance salt tolerance [27]. A statistical model that incorporates morphological, physiological, and biochemical factors would be more appropriate [28]. As a result, multivariate analysis helps to find the genetic origins of variation and to differentiate salt tolerance using several selection criteria. Methods for determining the salt tolerance of many genotypes must be economical, rapid, and easily quantifiable to achieve this goal [29]. Because of the variability in environmental variables from season to season, morphological measures sometimes need a large amount of phenotypic data and repetitive cropping seasons for a screening assessment. Variability in agricultural soil can also harm field evaluations, increasing the coefficient of variation, leading to breeders being led away from attaining their objectives [30]. Under salt stress, performing the morphological, physiological, and biochemical criteria would naturally differ among faba bean genotypes, with one genotype being superior in at least one feature while being poorer in others. Our research hypothesis was that the different faba bean genotypes would respond differently under different salinity conditions, modifying the cultivar tolerance ranking seen at different salt stress levels. This work aimed to investigate these discrepancies to see if these measures might be used as reliable screening criteria for genotype assessment in salinity circumstances with advanced statistical techniques.

## 2. Materials and Methods

### 2.1. Plant Materials and Experimental Set-Up

Twelve faba bean genotypes (*Vicia faba* L.) were used in this study (Table 1). A pot experiment (30 cm height × 20 cm width) was conducted to examine the efficiency of the multivariable morpho-agro-physiological traits to estimate the salt tolerance of the tested genotypes under t greenhouse conditions at the College of Food and Agriculture Sciences, Riyadh, Saudi Arabia. The average temperature was ±20 during the day and ±16 °C during the night. The photoperiodic management for the greenhouse system was a 16 h light and 8 h dark cycle. Humidity was maintained at around 60–70%. The pots were filled with pure sand that was irrigated twice a week with tap water along with 1/10th strength Hoagland nutrient solution. Three seeds were sown in each pot, and after the establishment of a seedling, two seedlings were maintained under controlled and stress conditions and then averaged. The genotypes were evaluated under three salinity levels (control, 100, and 200 mM NaCl). The seeds of each genotype were allowed to grow for 15 days to establish the seedlings before exposing it to salt stress. The seedlings were gradually subjected to salt stress starting from 100 mM for two weeks to avoid osmotic shock. After attaining complete stress levels, different morpho-physiological traits were measured using standard protocols. The data for stomatal conductance mmol m$^{-2}$ s$^{-1}$ (SC), leaf temperature (RT), and SPAD readings were collected at the vegetative stage. For the ionic concentration (Na$^+$, K$^+$) and total chlorophyll content (µg/mL), plant samples were collected in triplicate at the seventh fully leaf expanded stage. The experiment was a split-plot arrangement with an RCBD design (the main plot assigned the salinity treatments, and the genotypes assigned the subplot). The selected growth attributes were measured using the mean value of two plants or samples of uniform growth per factor, genotype, and three replications. For all of the parameters tested, relative trait changes (RTC) were computed as (CD)/C.

**Table 1.** Name and the sources of genotypes used in the study.

| Sr# | Genotype Name | Source |
|---|---|---|
| 1 | Hassawi-1 | Saudi Arabia |
| 2 | Gazira | Sudan |
| 3 | Hassawi-3 | Saudi Arabia |
| 4 | Triple white | Sudan |
| 5 | Sakha | Egypt |
| 6 | ILB-4347 | ICARADA |
| 7 | Misr-3 | Egypt |
| 8 | FLIP12501FB | ICARADA |
| 9 | Sakha-1 | Egypt |
| 10 | Hassawi-2 | Saudi Arabia |
| 11 | FLIP12504FB | ICARADA |
| 12 | FLIP12505FB | ICARADA |

### 2.2. Measurement of Growth Parameters

Leaf Fresh weight (g plant$^{-1}$), leaf dry weight (g plant$^{-1}$), plant height at maturity (cm), and leaf area (cm$^2$) were measured using a portable leaf area meter (Li-3000C). Days to 50% flowering, days to maturity, the number of pods per plant, the number of seeds per pod, the 100 seed weight (g), and seed yield (g plant$^{-1}$) were measured as well.

The relative water content (RWC), water deficit (WD), and relative turgidity (RT) were determined according to the methods determined by Grzesiak et al. [31] and Weatherley [32]. Fresh leaves (5 cm long) were used, which were then weighed. The fresh leaf weight was determined before the leaves were soaked in 100 mL of distilled water for 4 h.

The leaves' turgid weight (TW) was then measured. The same samples were then dried in the oven at 70 °C for 48 h to achieve the dry weight (DW). The measurements were used to calculate the following:

$$RWC = FW - DW/FW; RT = FW - DW/TW - DW; WD = 100 - RT$$

For the total chlorophyll (µg/mL) content, a leaf sample (0.1 g) was cut and put in 3 mL of methanol to determine the total chlorophyll content (ChL). The ChL contents were identified using TChL = $25.8 \times A\,650 + 4.0 \times A665$. The absorbance (650 and 665) was measured using a spectrophotometer. The chlorophyll was then converted to micrograms per gram of leaf tissue. The total chlorophyll ChL = (µg chlorophyll/mL methanol) $\times$ 3 mL methanol/(g tissue) was used to calculate the total chlorophyll [33]. Stomatal conductance and leaf temperature were determined using a leaf promotor. The SPAD reading was determined using a SPAD-502 meter.

A 0.3 g tissue subsample was placed in digestion tubes with 2 m/L concentrated sulfuric acid and left to sit for 15 min before being added to 2 mL of 30% hydrogen peroxide. The tubes were heated to 350 °C for 30 min and were then allowed to cool before adding 0.5 mL of 30% hydrogen peroxide. The method was repeated until a clear solution was obtained. The solution was then filtered and analyzed for the elements [34] and the concentration of the $Na^+$ and $K^+$ contents, which were determined using an EI microprocessor flame photometer model (1382). The $Na^+/K^+$ ratio was calculated by dividing the $Na^+$ content by the $K^+$ content.

Tolerance indices were used to determine the tolerance indices in the faba bean samples: yield stability index (YSI) = $Ys - Yc$ [35]; yield index (YI) = $\frac{Ys}{\overline{Ys}}$ [36]; salinity susceptible index (SSI) = $[1 - \left(\frac{Ys}{Yc}\right)]/[1 - (\frac{\overline{Ys}}{\overline{Yc}})]$, according to Fischer and Maurer [37]; salinity tolerance index (STI) = $Ys x \frac{Yc}{(\overline{Yc})^2}$ [38]; tolerance index (TOL) = $(Yc - Ys)$ [39]; mean productivity index (MPI) = $(Ys + Yc)/2$ [39]; relative efficiency index (REI) = $\frac{Ys}{\overline{Ys}} \times \frac{Yc}{\overline{Yc}}$ [40].

$\overline{Ys}$ = average yield of all genotypes under stress conditions. $\overline{Yc}$ = average yield of all genotypes under controlled conditions. Ys = average of individual genotypes under stress. Yc = average of individual genotype yield under control condition.

### 2.3. Statistical Analysis

The data were subjected to ANOVA using the SAS 9.0 software, and means were compared using LSD at a 5% probability level. The descriptive statistics were analyzed using the PAST 3.11 software, and Pearson's correlation coefficients were calculated the XLSTAT statistical package (Version 2018, Excel Add-ins soft SARL, New York, NY, USA). The mean data from the field were analyzed using the Euclidian distance. These distances were used to construct a dendrogram using the unweighted pair-group method with an arithmetic average (UPGMA) by employing the PAST (version 3.11) program [41]. The qualitative data were standardized using data transformation techniques and by analyzing the data and making clusters based on the Euclidean distance to study the similarity among the group of genotypes. The values of the indices were used again to analyze the PCA, construct biplot, treatments, and trait loading data of faba bean genotypes.

### 3. Results

The descriptive statistics of the studied characters are presented in Table 2. The table includes minimum, maximum, mean, and standard deviation (SD) values. All parameters recorded a high range of difference among the studied characteristics, demonstrating that salinity affects the agronomic, physiological, and biochemical characteristics in faba bean genotypes (Table 2). The mean average trait values of all the faba bean genotypes are presented in the Table S1 in the Supplementary Materials.

**Table 2.** Descriptive statistics of the studied variables.

| Variable | Minimum | Maximum | Mean | SD |
|---|---|---|---|---|
| Plant height (PH), | 14.66 | 47.00 | 28.89 | 5.80 |
| Leaf area (LA), | 7.90 | 31.74 | 15.69 | 3.89 |
| SPAD | 20.93 | 42.66 | 32.62 | 4.46 |
| Stomatal conductance mmol ($m^{-2}\,s^{-1}$) | 218.60 | 616.99 | 378.83 | 87.71 |
| Leaf temperature (LT) | 28.33 | 33.00 | 31.42 | 1.20 |
| Days to 50% flowering (DF) | 28.66 | 53.00 | 46.14 | 4.66 |
| Fresh weight (g) (FW) | 0.22 | 0.89 | 0.49 | 0.13 |
| Dry weight (g) DW | 0.008 | 0.17 | 0.05 | 0.03 |
| Number of pods (NOP) | 6.33 | 70.33 | 27.63 | 15.10 |
| Number of seeds (NOS) | 9.66 | 187.66 | 63.68 | 38.44 |
| Seed weight (g) (SW) | 5.40 | 73.39 | 26.32 | 16.08 |
| $Na^+$ | 6.60 | 192.00 | 57.25 | 50.18 |
| $K^+$ | 7.40 | 148.00 | 49.20 | 30.57 |
| $Na^+/K^+$ | 0.117 | 6.52 | 1.46 | 1.44 |

The analysis of variance data is presented in Table 3. All of the studied characteristics recorded significant differences among the control and salinity treatments.

**Table 3.** Analysis of variance (ANOVA).

| SOV | DF | PH | LA | SPAD | SC | LT | FD | FW | DW | NOP | NOS | $Na^+$ | $K^+$ | $Na/K^+$ | T.ChlL | SW |
|---|---|---|---|---|---|---|---|---|---|---|---|---|---|---|---|---|
| Salinity | 2 | 207.38 | 45.79 | 266.64 | 41,025.2 | 24.57 | 185.40 | 0.062 | 0.001 | 3252.86 | 20,052.0 | 20,855.9 | 2802.69 | 17.64 | 1943.46 | 3279.25 |
| Variety | 11 | 17.24 | 66.20 | 84.81 | 39,415.4 | 3.08 | 133.81 | 0.070 | 0.002 | 141.47 | 1794.1 | 2533.9 | 771.46 | 2.53 | 68.49 | 388.73 |
| S × V | 22 | 60.19 | 48.29 | 66.08 | 6246.9 | 2.84 | 67.66 | 0.011 | 0.0008 | 236.42 | 955.8 | 883.0 | 989.25 | 0.81 | 46.85 | 163.40 |
| Error | 70 | 28.10 | 36.93 | 56.60 | 2115.8 | 1.91 | 16.21 | 0.009 | 0.0004 | 114.98 | 856.0 | 1066.9 | 603.65 | 1.24 | 117.56 | 190.54 |

Plant height (PH), leaf area (LA), stomatal conductance (SC), leaf temperature (LT), days to 50% flowering (FD), fresh weight (g) (FW), dry weight (g) DW, number of pods (NOP), number of seeds (NOS), seed weight (g) (SW), Total chlorophyll μg/mL (TChL).

The principal components analysis of all of the studied traits based on the combined (control, S1 = 100 mM and S2 = 200 mM) treatments were analyzed (Table 4). The first three PCs represent 81.45% of the total variance among the studied traits.

**Table 4.** The loading traits for the first 3 PCs.

| Trait | PC 1% Variance = 64.77 | PC 2% Variance = 10.91 | PC 3% Variance = 5.76 |
|---|---|---|---|
| Plant height (PH), | 0.05 | 0.04 | −0.02 |
| Leaf area (LA), | 0.01 | 0.07 | 0.22 |
| SPAD | 0.01 | −0.04 | 0.07 |
| Stomatal conductance (SC), | 0.04 | 0.04 | 0.04 |
| Leaf temperature (LT) | −0.01 | 0.02 | 0.00 |
| Days to 50% flowering (DF) | 0.03 | 0.01 | 0.03 |
| Fresh weight (g) (FW) | −0.07 | 0.09 | 0.29 |
| Dry weight (g) DW | −0.09 | 0.24 | 0.77 |
| Number of pods (NOP) | 0.24 | −0.10 | −0.05 |
| Number of seeds (NOS) | 0.28 | −0.11 | −0.28 |
| Seed weight (g) (SW) | 0.36 | 0.18 | −0.05 |
| Sodium ($Na^+$) | −0.51 | 0.49 | −0.26 |
| Potassium ($K^+$) | 0.09 | 0.73 | −0.24 |
| $Na^+/K^+$ ratio | −0.60 | −0.24 | −0.02 |
| Total chlorophyll (TChL) | 0.31 | 0.19 | 0.24 |

The biplot analysis (PC1 and PC2) was presented in Figure 1. The results suggested that the genotypes (Hassawi-2, Sakha, ILB-4347, Misr-3, FLIP12501FB) were present in the positive region of PC1 (first quadrant) under control conditions. Similarly, Hassawi-1,

Hassawi-2, Sakha, Gazira, and FLIP12504FB were present in the same group. The most notable characteristics that contributed to the variance under control and salinity conditions were the $K^+$, seed weight (g), and days to 50% flowering. The genotypes (Sakha-1, triple white, Hassawi-3, Gazira, Hassawi-3) were present in PC1 (second quadrant). The significant characters contributed more in PC1 (2nd quadrate), SPAD, seed number, and pods/plant (Figure 1). However, at higher salinities (S2 = 200 mM), the more significant genotypes are (Hassawi-2, Hassawi-1, ILB-4347) in one group; however, FLIP12501FB, Sakha-1, and triple white are away from the center (sensitive) but are present in PC2 (second quadrant) (Figure 1). The most important characteristics contributed more in terms of PC2 leaf fresh weight (g), leaf dry weight (g), leaf area (cm$^2$), Na$^+$, K$^+$, and Na$^+$/K$^+$).

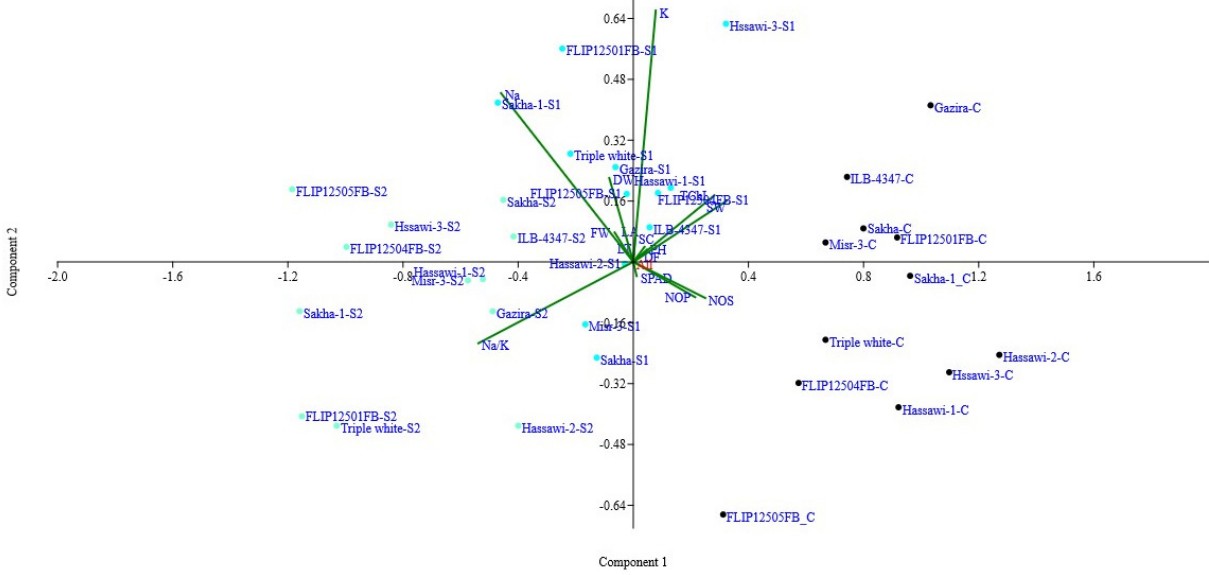

**Figure 1.** Biplot analysis (PC1 and PC2) of combined data (control and salinity treatments) and genotype names.

We generated the regression analysis to determine the prediction of the variables studied (Table 5). The table describes the slope, error, intercept, correlation coefficient, and probability values. The regression results suggested that the characteristics, i.e., the number of pods/plant, number of seeds/pod, seed weight, and Na$^+$/K$^+$ ratio, were more affected by an increase in salinity. The leaf area (cm$^2$) remained unaffected as the salinity increased (Table 5).

**Table 5.** Regression analysis of the studied parameters (agronomic, physiological, biochemical) under salinity conditions.

| Variable | Slope | Error | Intercept | Error | r | p |
|---|---|---|---|---|---|---|
| Leaf area (cm$^2$) (LA) | $2.52 \times 10^{-5}$ | 0.12 | 17.2 | 3.77 | $3.58 \times 10^{-5}$ | 1.00 |
| SPAD (GLI) | $-0.20$ | 0.14 | 38.2 | 4.24 | $-0.25$ | 0.14 |
| Stomatal conductance mmol (m$^{-2}$ s$^{-1}$) (SC) | 2.02 | 2.85 | 339.4 | 89.16 | 0.12 | 0.48 |
| Leaf temperature (LT) | $-0.05$ | 0.03 | 33.0 | 1.02 | $-0.25$ | 0.14 |
| Days to 50% flowering (DF) | 0.15 | 0.12 | 38.5 | 3.64 | 0.21 | 0.22 |
| Fresh weight (g) (FW) | 0.00 | 0.00 | 0.7 | 0.12 | $-0.18$ | 0.30 |
| Dry weight (g) DW | 0.00 | 0.00 | 0.1 | 0.02 | $-0.22$ | 0.21 |
| Number of pods (NOP) | 1.04 | 0.35 | $-6.1$ | 10.93 | 0.46 | 0.01 |
| Number of seeds (NOS) | 2.29 | 0.93 | $-11.3$ | 29.08 | 0.39 | 0.02 |
| Seed weight (g) (SW) | 1.33 | 0.41 | $-16.1$ | 12.81 | 0.49 | 0.00 |
| Sodium (Na$^+$) | $-1.73$ | 1.14 | 103.0 | 35.57 | $-0.25$ | 0.14 |
| Potassium (K$^+$) | 0.80 | 0.81 | 25.3 | 25.35 | 0.17 | 0.33 |
| Na$^+$/K$^+$ ratio | $-0.08$ | 0.03 | 3.7 | 1.08 | $-0.37$ | 0.03 |
| Total chlorophyll (T.ChL) | 0.58 | 0.35 | 8.0 | 10.83 | 0.28 | 0.10 |

The Pearson's correlation matrix was also determined for the studied characteristics using the combined mean data of the control, S1 = 100 mM, and S2 = 200 mM (Figure 2).

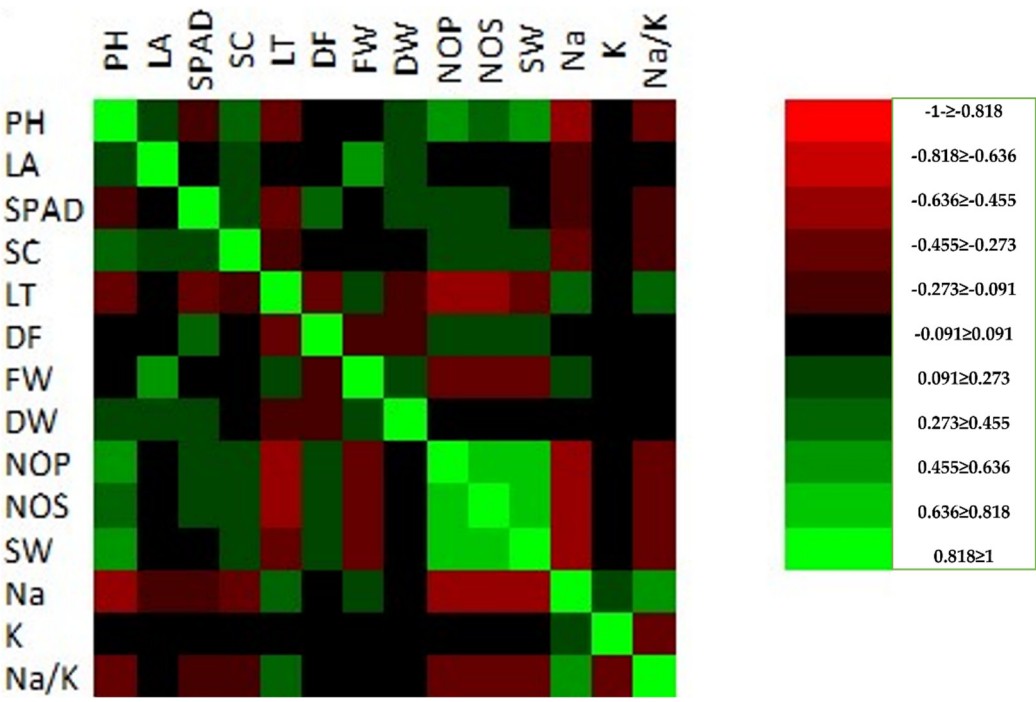

**Figure 2.** Pearson's correlation matrix of the studied characteristics under control and salinity (S1 = 100 mM, S2 = 200 mM) conditions (measuring scale on the right side represents highly insignificant (light red) to highly significant correlations (light green) at a ≥5% probability level. Plant height (PH), leaf area (LA), stomatal conductance mmol $m^{-2}$ $s^{-1}$ (SC), leaf temperature (LT), days to 50% flowering (DF), fresh weight (g) (FW), dry weight (g) DW, number of pods (NOP), number of seeds (NOS), seed weight (g) (SW), total chlorophyll μg/mL (T. ChL).

A significant correlation was recorded for plant height (cm) with leaf area ($cm^2$), stomatal conductance mmol ($m^{-2}$ $s^{-1}$), dry weight (g), number of pods, number of seeds, and seeds weight (g), while a negative correlation was recorded for SPAD, leaf temperature, $Na^+$, $K^+$, and $Na^+/K^+$. The fresh weight (FW) and dry weight (DW) were the characteristics that were the most significantly affected by the different salinity conditions. The FW was strong negatively correlated to the PH, SPAD, SC, DF, NOP, NOS, SW, K, and $Na^+/K^+$. The absorbance of the $Na^+$, $K^+$, and $Na^+/K^+$ indicates the salinity tolerance under the field conditions. These three parameters ($Na^+$, $K^+$, and $Na^+/K^+$) were recorded as being insignificant for all of the studied parameters; however, a significant correlation was recorded for LT and FW (Figure 2). The cluster analysis was used to group the faba bean genotypes based on the control, 100 mM, and 200 mM salt stress conditions (Figure 3a–c). The cluster analysis for control the treatments was divided into two groups. The first group was further subdivided into two subgroups. The first group compassed the 11 genotypes, while the FLIP12505FB genotype was individually separated (Figure 3c). Under the 100 mM condition, the genotypes were equally distributed between the two main groups. Each main group was subdivided into two subgroups under the 100 mM salt stress condition. Sakha, Misr-3, Hassawi-1, Gazira, Hassawi-2, and ILB-4347 were allocated to group I (Figure 3b); however, triple white was grouped with Sakha-1, Hassawi-3, and three genotypes of FLIP1250FB. Similarly, under the 200 mM salt stress condition, the genotypes were clustered into two main groups, and the Gazira genotype failed to be grouped and was separated into a group of its own. Group 1 comprised five genotypes (FLIP12505FB, triple white, FLIP12504FB, FLIP12501FB, and Sakha-1), while group II contained the Hassawi-1, Hassawi-2, Hassawi-3, Sakha, and Misr-3 genotypes (Figure 3c).

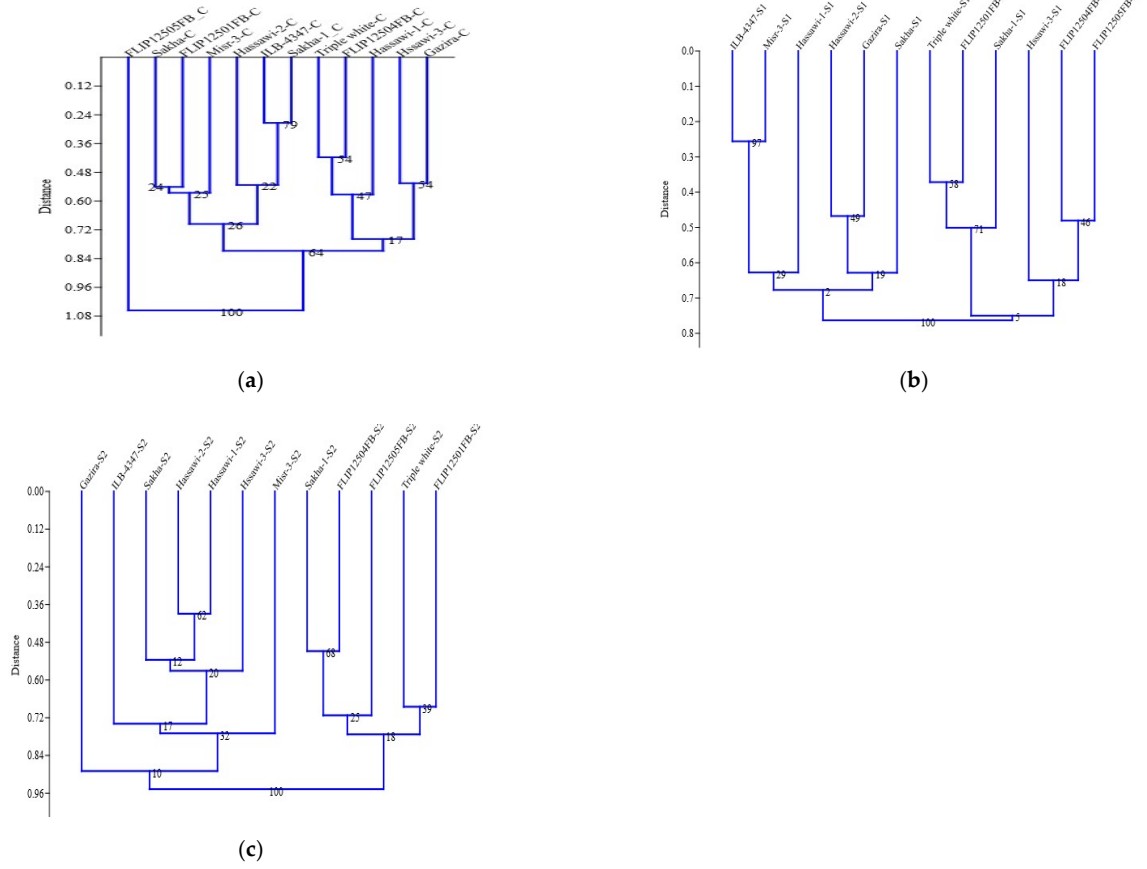

**Figure 3.** (**a**–**c**) Cluster analysis based on control (**a**), (**b**) S1 (100 mM), and (**c**) S2 (200 mM) mean data.

Some salt tolerance indexes (YI, SSI, STI, TOL, MPI, MRP, REI) were also calculated based on seed weight/plant to determine the susceptibility and tolerance level of each faba bean genotype (Table 6).

**Table 6.** Index (YI, SSI, STI, TOL, MPI, REI) values for all studied faba bean genotypes.

| Genotypes | YI 100 mM | YI 200 mM | SSI 100 mM | SSI 200 mM | STI 100 mM | STI 200 mM | TOL 100 mM | TOL 200 mM | MPI 100 mM | MPI 200 mM | REI 100 mM | REI 200 mM |
|---|---|---|---|---|---|---|---|---|---|---|---|---|
| Hassawi-1 | 1.13 | 0.91 | 0.800 | 0.596 | 1.145 | 0.257 | 17.20 | 35.67 | 42.58 | 26.93 | 0.843 | 0.926 |
| Gazira | 1.64 | 0.97 | 0.241 | 0.574 | 0.807 | 0.277 | 4.48 | 35.78 | 36.72 | 27.49 | 1.299 | 1.107 |
| Hssawi-3 | 1.08 | 1.23 | 0.357 | 0.379 | 0.665 | 0.304 | 12.64 | 27.13 | 33.02 | 25.77 | 1.070 | 1.215 |
| Triple white | 1.03 | 1.08 | 0.512 | 0.365 | 0.544 | 0.229 | 8.23 | 23.02 | 29.62 | 22.22 | 0.877 | 0.914 |
| Sakha | 0.98 | 1.66 | 0.744 | −0.191 | 0.424 | 0.290 | 3.54 | 11.22 | 25.95 | 22.11 | 0.683 | 1.158 |
| ILB-4347 | 0.82 | 1.25 | 0.429 | 0.646 | 0.898 | 0.557 | 50.40 | 58.04 | 45.32 | 41.49 | 1.446 | 2.226 |
| Misr-3 | 0.86 | 1.07 | 0.352 | 0.324 | 0.426 | 0.213 | 10.23 | 20.89 | 26.45 | 21.11 | 0.686 | 0.852 |
| FLIP12501FB | 1.16 | 0.78 | 0.465 | 0.604 | 0.705 | 0.190 | 10.43 | 31.29 | 33.78 | 23.36 | 1.136 | 0.761 |
| Sakha-1 | 0.71 | 0.60 | 0.045 | 0.645 | 0.372 | 0.127 | 16.02 | 27.58 | 25.53 | 19.75 | 0.599 | 0.506 |
| Hassawi-2 | 1.46 | 1.16 | 0.016 | 0.185 | 1.678 | 0.537 | 37.28 | 61.83 | 54.75 | 42.47 | 2.702 | 2.146 |
| FLIP12504FB | 0.75 | 0.54 | 0.520 | 0.559 | 0.289 | 0.084 | 5.88 | 19.11 | 21.57 | 14.96 | 0.466 | 0.335 |
| FLIP12505FB | 0.37 | 0.75 | 0.341 | −0.099 | 0.077 | 0.063 | 4.45 | 6.08 | 11.28 | 10.46 | 0.125 | 0.254 |

Yield index (YI), salinity susceptible index (SSI), salinity tolerance index (STI), tolerance index (TOL), mean productivity index (MPI), relative efficiency index (REI).

The maximum YI was reported in Gazira, followed by Hassawi-2 under the 100 mM salinity condition. The minimum YI was recorded in the FLIP12505FB genotype. Similarly, the maximum YI was observed in Sakha followed by in ILB-4347 (1.25), Hassawi-3 (1.23), and Hassawi-2 (1.16 under the 200 mM salt stress condition. The minimum YI (0.54) was

recorded (FLIP12504FB) genotype under a 200 mM salt stress level. The maximum SSI (0.800) was recorded in Hassawi-1 (100 mM), ILB-4347 (0.646; 200 mM), while the minimum SSI (0.016) was recorded for Hassawi-2 under the 100 mM salt stress condition. The maximum (1.678) salinity tolerance index was recorded in Hassawi-2, and the minimum was recorded in the FLIP12504FB faba bean genotype under the 100 mM salinity conditions, while for the 200 mM salinity conditions, the maximum was recorded for ILB-4347 and Hassawi-2, respectively. The mean productivity index was recorded maximum for Hassawi-2 under both the 100 mM and 200 mM salt stress conditions, while the minimum was recorded for the FLIP12505FB genotype under both stress conditions. The maximum relative efficiency index was recorded in Hassawi-2 followed by in ILB-4347 under both of the salt stress (100 mM and 200 mM) conditions. The correlation matrix was also determined among the salt stress indices under moderate and high salt (100 mM and 200 mM) stress conditions (Table 7). The YI recorded a negative correlation with SSI at both stress levels; however, all of the other index (SSI, STI, TOL, MPI, REI) values recorded positive and significant correlations.

**Table 7.** Pearson's correlation values are based on stress indices ($p = 5\%$) for all faba bean genotypes.

| Genotypes | YI 100 mM | YI 200 mM | SSI 100 mM | SSI 200 mM | STI 100 mM | STI 200 mM | TOL 100 mM | TOL 200 mM | MPI 100 mM | MPI 200 mM | REI 100 mM | REI 200 mM |
|---|---|---|---|---|---|---|---|---|---|---|---|---|
| YI (100 mM) | | 0.38934 | 0.446 | 0.1394 | 0.0007 | 0.1110 | 0.477 | 0.066 | 0.002 | 0.0483 | 0.0007 | 0.111 |
| YI (200 mM) | 0.274 | | 0.990 | 0.1886 | 0.3772 | 0.0250 | 0.490 | 0.633 | 0.276 | 0.1225 | 0.3774 | 0.025 |
| SSI (100 mM) | −0.243 | −0.004 | | 0.1525 | 0.3994 | 0.0574 | 0.003 | 0.020 | 0.220 | 0.0452 | 0.3993 | 0.057 |
| SSI (200 mM) | 0.453 | −0.408 | 0.440 | | 0.0488 | 0.2955 | 0.081 | 0.005 | 0.023 | 0.0737 | 0.0488 | 0.296 |
| STI (100 mM) | 0.836 | 0.281 | 0.268 | 0.579 | | 0.0024 | 0.023 | 0.000 | 0.000 | 0.0002 | 0.0000 | 0.002 |
| STI (200 mM) | 0.484 | 0.640 | 0.562 | 0.330 | 0.787 | | 0.001 | 0.001 | 0.000 | 0.0000 | 0.0024 | 0.000 |
| TOL (100 mM) | 0.227 | 0.221 | 0.771 | 0.523 | 0.649 | 0.829 | | 0.000 | 0.006 | 0.0004 | 0.0225 | 0.001 |
| TOL (200 mM) | 0.547 | 0.154 | 0.658 | 0.750 | 0.872 | 0.847 | 0.874 | | 0.000 | 0.0000 | 0.0002 | 0.001 |
| MPI (100 mM) | 0.790 | 0.342 | 0.382 | 0.647 | 0.964 | 0.867 | 0.738 | 0.934 | | 0.0000 | 0.0000 | 0.000 |
| MPI (200 mM) | 0.580 | 0.471 | 0.586 | 0.534 | 0.873 | 0.970 | 0.855 | 0.944 | 0.948 | | 0.0002 | 0.000 |
| REI (100 mM) | 0.836 | 0.280 | 0.268 | 0.578 | 1.000 | 0.787 | 0.649 | 0.872 | 0.964 | 0.87247 | | 0.002 |
| REI (200 mM) | 0.483 | 0.640 | 0.562 | 0.329 | 0.787 | 1.000 | 0.829 | 0.847 | 0.867 | 0.96968 | 0.7872 | |

Yield index (YI), salinity susceptible index (SSI), salinity tolerance index (STI), tolerance index (TOL), mean productivity index (MPI), relative efficiency index (REI).

## 4. Discussion

The present study was conducted to determine the tolerance of different faba bean genotypes using different agro-morphological, physiological, and biochemical characteristics under different salinity levels. The descriptive and analysis of variance values recorded significant differences among the studied characteristics. Similar results [42] suggested that significant differences were recorded in the studied characteristics under different abiotic stress conditions in barley varieties. Among different types of legumes, Faba bean is salt-sensitive a [43]. In the faba bean genotypes studied here, the salt treatments lowered biomass production and water intake [44]. Ahmad et al. [43] also found that NaCl inhibited faba bean growth and biomass yield. Cell division inhibition and cell elongation are the result of the loss in growth and biomass yield caused by salinity [45]. NaCl reduces growth and biomass yield by reducing mineral intake, generating reactive oxygen species, inhibiting enzyme activity, and causing hormonal imbalances [46]. Ahmad et al. [43] suggested that sensitive cultivars are more vulnerable to damage relative to tolerance in pea plants and mulberry seedlings [43]. Similar results were reported in *Solanum lycopersicum* and *Vicia faba* [42]. Sodium and potassium ions share similar physiochemical structures and compete for Na and K ion uptake in the soil [47]. Other plant species, such as mustard [48] and strawberry, have increased $Na^+$ buildup with lower $K^+$ and $Ca^+$ absorption when exposed to NaCl [49]. The inhibition of these mineral elements, the primary cause of stunted plant growth and development, is caused by high saline levels [47].

The analysis of variance results recorded high levels of significant differences among the studied traits under control and salinity. Similar results were reported by Filipović et al. [50] and suggested that increasing the salinity (100 mM) of the irrigation water significantly reduced morphological characteristics. The biplot loadings for PC1 and PC2 were performed related to salinity in order to group the faba bean genotypes. Similar results were recorded when phenotypical data were analyzed, and the genotypes were grouped in the hierarchical cluster pattern [51]. Multivariate analytical techniques such as PCA and cluster analysis were used [52] and was significant in determining the best genotypes under stress conditions. The findings refer to Karadavut [53] and Mekonnen et al. [54]. Similarly, the cluster analysis results reported that the faba bean genotypes were divided into two groups under the control and 100 mM salinity levels; however, the 200 mM salinity l level recorded three groups of faba bean genotypes, showing that salinity stress may limit phenotypic variability among faba bean genotypes. The genotypic variation was decreased, and the genotype distribution in the group was also different because of the increased salt stress level. Similar results were in line with those recorded by Saed-Moucheshi et al. [55], who suggested that stress reduced the phenotypic variation among the studied variables and limited the diversity potential in crop plants, such as in triticale [56,57].

Some essential multivariate strategies have been adequately employed to determine tolerant and susceptible genotypes in response to varied environmental situations. In-plant breeding and screening programs, clustering analysis, regression approaches, and principal component analysis are popular methodologies [58]. There are a variety of univariate strategies for detecting tolerant genotypes that work in different ways. Other studies have identified a variety of univariate methodologies that are suitable for determining the optimum genotype for various environmental situations. The susceptibility and tolerant indices are powerful tools that help to screen genotypes under abiotic stress conditions [55]; similarly, STI is also a vital tolerance index to determine the relationship among the other stress indices [59]. A positive correlation was found with TOL under control conditions in our study. However, a negative correlation was reported under stress conditions. Similar results were reported by Talebi et al. [60]. Similar results were also reported in another study, suggesting that there is positive association between TOL under control conditions and a negative association under stress conditions suggests that TOL selection would reduce yield under control conditions [61,62]. As a result, integrating multivariate approaches with tolerance indices is an efficient way to use all of the computed indices in genotype screening systems. Besides our study, Fernandez [63] used the multivariate analysis technique in different crop plants such as mung bean and wheat [60,64,65], and used a similar type of analysis [62] in lentil to introduce tolerant and susceptible genotypes. This tolerance could be because of genetic variances in salinity tolerance rather than differences in ROS detoxification capabilities. Differences in antioxidant enzyme expression or activity are linked to more tolerant genotypes, but they can also be linked to more sensitive genotypes. The differences among agro-morphological, physiological, and biochemical characteristics among faba bean genotypes could be because of genotypic differences in stomatal closure or in other responses that affect the rate of $CO_2$ fixation [10].

## 5. Conclusions

The study confirms the significance of the studied traits, salinity index, and their association to determine the salinity tolerance of each genotype. The presentation of the data using multivariate analyses (descriptive, ANOVA, PCA, cluster analysis, regression analysis, correlation analysis, and salinity indices) also forces the establishment and definition of tolerance levels. Four genotypes, Hassawi-2, ILB-4347, Sakha, Misr-3, and Flip12501FB, showed significant tolerance levels; however, FLIP12504FB represents a sensitive genotype based on its final grain yield. Significant index correlations with grain yield supports the use of these indices as tools for screening faba bean genotypes for salinity stress evaluation. In short, these findings show that faba bean genotypes can perform in a semiarid environment and can be grown under salt stress conditions where high temperatures and

salinity are the constraints. The salinity tolerance trait is complicated and can be easily improved through breeding selection. However, the best results could be achieved through molecular genetics and genomics techniques by unraveling the molecular mechanism involved when crops are under salinity stress conditions. Furthermore, characterizing new salt tolerance genes could help researches to improve the faba bean genome under abiotic stress conditions in the future, especially under high salinity and high temperature conditions. Consider these tolerant genotypes and utilize them for breeding and genetic programs to improve salinity sensitivity in faba bean.

**Supplementary Materials:** The following supporting information can be downloaded at: https://www.mdpi.com/article/10.3390/agriculture12020235/s1, Table S1: Mean average trait values of all the faba bean genotypes.

**Author Contributions:** Conceptualization, M.A and H.H.M.; Data curation, S.A.A.-F.; Formal analysis, M.A.; Investigation, E.E.-H.; Methodology, E.E.-H.; Resources, S.S.A.; Supervision, S.S.A.; Writing—original draft, M.A.; Writing—review & editing, M.A. and H.H.M. All authors have read and agreed to the published version of the manuscript.

**Funding:** This research was funded by the Deanship of Scientific Research in King Saud University, the initiative of the DSR Graduate Students Research Support (GSR).

**Institutional Review Board Statement:** Not applicable.

**Informed Consent Statement:** Not applicable.

**Data Availability Statement:** The corresponding authors can provide the data used in this study upon request.

**Conflicts of Interest:** The authors declare no conflict of interest.

**Abbreviations**

Analysis of variance (ANOVA); days to flowering (DF); leaf dry weight (DW); leaf fresh weight (FW); green leaf index (GLI); potassium ($K^+$); leaf area ($cm^2$) (LA); leaf temperature (LT); mean productivity index (MPI); sodium ($Na^+$); number of pods/plant (NOP); number of seeds (NOS); principal component analysis (PCA); plant height (PH); relative efficiency index (REI); relative turgor (RT); relative water contents (RWC); statistical analysis system (SAS); stomatal conductance (SC); standard deviation (SD); soil–plant analysis development (SPAD); salinity susceptible index (SSI); salinity tolerance index (STI); source of variation (SOV); seeds weight (SW); total chlorophyll content (T. Chl); tolerance index (TOL); turgor weight (TW); unweighted pair group method with an arithmetic average (UPGMA); water deficit (WD); yield index (YI)

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
