# Peer review of "Agronomical and Physiological Responses of Faba Bean Genotypes to Salt Stress"

_agriculture, doi:10.3390/agriculture12020235_

Round 1
Reviewer 1 Report
Major comments:
- Abstract: it is impossible to understand the abstract without deciphering the abbreviation
- The work should contain all experimental data for each genotype tested either in the text or at least in Supplement file
- The text of the abstract should be revised so that the results should be stated more clearly. See below:
‘The most significant characters contributed more to the diversity was (PH, LA, SPAD, SC, 25 DF, NoP, NOS, SW, K, and TChL); however, the significant genotypes (Hassawi-2, Sakha, ILB-4347, Misr-3, FLIP12501FB) were present in the PC1 under both conditions. The results predicted that the Hassawi-2, ILB-4347, Gazira, and Flip12501FB were the significant (tolerant) genotypes. However, FLIP1250FB represents a sensitive one based on 29 final grain yield’
So: Is FLIP1250FB genotype tolerant or sensitive??? -if Sakha and Misr-3 are significant why are they absent in the list of significant (tolerant) group?- or it is a misprint? And these are just repetitions????
- In Materials and Methods section it is necessary to indicate the method of Na, K determination- citing the reference is not enough
- Salinity usually causes great changes in protein and carbohydrate content. Please explain why you did not use these parameters
- Due to enormous amount of abbreviations it is highly desirable not only to decipher each one in the text and in footnotes of Tables and in Figure legends but also to add a special abbreviation list either at the beginning or at the end of the manuscript
Minor comments:
There are some misprints in the text:
1)line 23 ‘an insignificant association was recorded LT, FW’- change to ‘an insignificant association was recorded with LT, FW’
2)line 25 ‘The most significant characters contributed more to the diversity was (PH, LA, SPAD, SC, DF, NoP, NOS, SW, K, and TChL’ change ‘was’ to ‘were’ and abbreviations are to be deciphered; change ‘characters’ to ‘characteristics’
3)line 26 ‘however, the significant genotypes were (Hassawi-2, Sakha, ILB-4347, Misr-3, FLIP12501FB) were present in the PC1 under both conditions’- delete one ‘were’
4)line 40 ‘. Faba bean is an older crop of the world’- may be ‘. Faba bean is one of the oldest crops of the world’???
5)line 42 ‘Its age and commercial importance, faba bean is a diploid species//”- style???
6)line 77 ‘As a result, multivariate analysis helps find the genetic origins’ change to ‘As a result, multivariate analysis helps to find the genetic origins’
7) line 95’ ‘college of food and agriculture’ change to ‘College of Food and Agriculture’
8)Line 109 ‘cm2’ change to ‘cm2’
9) line 113 ‘and Weatherley [30] was used to determined’- style
10)Line 110 what is RT?
11)Line 127 ‘Ion concentrations of Na+, K+, and Na+/K+ were determined according to Yoshida et al. [33]’. Wrong citation [33}- there is nothing about the determination of Na+, K+. Please, add appropriate citation and write what method of determination was used
12)Line 129 ‘Tolerance indices were used to determine the tolerance indices in the fababean. Yield stability index (YSI) = Ys-Yc [34], Yield index (YI) =.Ys YÌ…s’- decipher what Ys and Yc are.
13)Line 130 please check formula; SSI, STI, TOL, MPI.
I can’t understand formula: 1 − ( ?? ?? )/1− ( ?? ?? ),your numerator and denominator are similar, then this ratio will be equal to 1
Salinity tolerance index (STI)= Ys) Yc (?Ì… ? ) 2, Tolerance index (TOL)= Yc − Ys [) 3], Mean productivity index (MPI)=(?? + ??)/) , relative efficiency index (REI)= ?? YÌ…s ? ?? – decipher abbreviations
14) 2- add units of the parameters. Can’t understand how Na/K ratio was calculated???
15)Table 3- what is SOV???
16)Reference list :
- should be revised according to the authors guidelines (use Journals abbreviations everywhere)
- References 29 and 27 duplicate each other, delete one
- To my opinion it is desirable to add some more citations of 2020-2021 either in Introduction or/and in the text:
- Yang, F., Chen, H., Liu, C. et al.Transcriptome profile analysis of two Vicia faba cultivars with contrasting salinity tolerance during seed germination. Sci Rep 10, 7250 (2020). https://doi.org/10.1038/s41598-020-64288-7
- Bimurzayev, N., Sari, H., Kurunc, A. et al.Effects of different salt sources and salinity levels on emergence and seedling growth of faba bean genotypes. Sci Rep 11, 18198 (2021). https://doi.org/10.1038/s41598-021-97810-6
- L FILIPOVIĆ, D ROMIĆ, G ONDRAŠEK, I MUSTAĆ, V FILIPOVIĆ The effects of irrigation water salinity level on faba bean (Vicia faba L.) productivity. Journal of Central European Agriculture, 2020, 21(3), p.537-542
Author Response
Dear professor
We appreciated the comprehensive suggestions and positive recommendations, which added value and improved our work. We try to reply to each point, and the correction is inserted into the manuscript using red color. The comments and responses are presented in the attached file.
Best regards

Reviewer 2 Report
The authors have presented a manuscript, describing the agronomical and physiological response of faba bean genotypes to salt stress. Following, I have included some comments to improve the manuscript.
- I suggest to the authors to add a new section detailing the state of the art. In this section, authors have to describe the relevant related work in which explain.
- Can the authors include at the end of the introduction, more details of the objectives of their study.
- Line 147: Table 2 Authors must also present.
- Line 151: Table 3 Authors must also present.
- This work presents very interesting results and practice to response of faba bean genotypes. I think that the authors can improve the format of results demonstration. The authors can highlight better the importance of the results obtained.
- Conclusions. Consider extending the conclusions and adding a Future works paragraph. The summary and Conclusions, it is better to combine them in only section of Conclusions.
- References: Line 387, 404, 417 and 480. Author must also present the change the year to bold and put the reference page.
Finally, the review is interesting and presents considerable information on the possibilities of the production of faba bean, but authors must improve the presentation of their results and discussion. The topic in interesting, but the study lacks more details with precision, and concrete conclusion that will help farmers to improve or to change their strategies in the agriculture.
Author Response
Dear Professor
Greetings
We appreciated the comprehensive suggestions and positive recommendations, which added value and improved our work. We try to reply to each point, and the correction is inserted into the manuscript using red color. The comments and responses are presented in the attached file.
Best regards

Reviewer 3 Report
Soil salinity is a serious problem, especially in certain geographic regions. Hence, the subject matter of the research presented in the manuscript is important and interesting. Manuscript requires corrections and additions, especially the chapters 'Material and methods' and 'Results'
Line [86] – ‘…wheat genotypes’ – for sure?
Line [92-98] - What substrate was used to fill the pots, in what quantity (weight), how many seeds were sown on one pot, how many plants were left. What temperature and humidity were maintained in the greenhouse, was the lighting regulated?
Line [108] - Shoot Fresh weight (g plant_1), shoot dry weight (g plant_1), - when was it marked. Table 2 shows a very small value of this feature (0.22-0.89 g and 0.008-0.17 g)
Line [109] - ‘…Flag leaf area’ - Is this really the correct term?
Line [119] - add an explanation of what WC, FW, RT means.
Line [120-121] - this sentence needs to be corrected.
When were the chlorophyll content and stomatal conductance measured? once during the growing season? what phase were the plants in?.
Line 128 - ions concetration in what?
Line [130]- add explanations to each of the calculated indices, what the letters in the equations mean, e.g. Ys, Yc, ….
Line [132] - There is something missing in the formula for the MPI index?
Line [147] – in table 2 not in table 1
Table 2 - What does SPAD mean? There is no information in the methodology that such an indicator was measured. The same is true for the leaf temperature. Why was no value provided for Days to maturity, for 100 - Seed Weight and for proline? ‘Fresh weight/ dry weight of what? Add missing units, e.g. for: PH, LA, LT, K+, Na+
Table 3 - Why is there no data for SW?
Which is shown in table 7. Pearson's correlation coefficients and p-value? If so, please explain below the table.
Line [271] - remove Ahmad et al.
Line [277-295] - this fragment needs to be corrected. The discussion is chaotic and superficial.
Author Response
DearProfessor
We appreciated the comprehensive suggestions and positive recommendations, which added value and improved our work. We try to reply to each point, and the correction is inserted into the manuscript using red color. The comments and responses are presented in the attached file.
Bst regards

Round 2
Reviewer 3 Report
Most of the comments included in the review were taken into account in the revised manuscript. I only have minor comments.
There is still no information on how and when the leaf temperature was measured.
Line [121-123] – ‘The selected growth attributes were measured using the mean value of three plants or samples of uniform growth per factor, genotype, and replication”. I do not understand. In how many repetitions the experiment was performed? In line [110-11] is ‘Three seeds were sown in each pot, after the establishment of seedling, two seedlings were maintained under controlled and stress conditions and then averaged’ So, there were 2 plants in one pot. So, for example, the height of two plants in one pot and one plant in the next was measured. This needs to be clarified. More plants should be considered for determining the morphological characteristics, i.e. plant height, number of pods, number of seeds etc.
Author Response
Please see the attachment
There is still no information on how and when the leaf temperature was measured.
Response: Line 116-118
Line 145-146
Line [121-123] – ‘The selected growth attributes were measured using the mean value of three plants or samples of uniform growth per factor, genotype, and replication”. I do not understand. In how many repetitions the experiment was performed? Inline [110-11] is ‘Three seeds were sown in each pot, after the establishment of seedling, two seedlings were maintained under controlled and stress conditions and then averaged’ So, there were 2 plants in one pot. So, for example, the height of two plants in one pot and one plant in the next was measured. This needs to be clarified. More plants should be considered for determining the morphological characteristics, i.e., plant height, number of pods, number of seeds, etc.
Response: Line 121-123